# Dietary sulfur amino acid restriction elicits a cold-like transcriptional response in inguinal but not epididymal white adipose tissue of male mice

Philip MM Ruppert[1]*, Aylin S Gueller[1], Marcus Skjæveland[1], Natasa Stanic[1], Jan-Wilhelm Kornfeld[1,2]*

[1]Functional Genomics and Metabolism Unit, Department for Biochemistry and Molecular Biology, University of Southern Denmark, Odense, Denmark; [2]Novo Nordisk Foundation Center for Adipocyte Signaling, University of Southern Denmark, Odense, Denmark

## eLife Assessment

The present study employed transcriptomics to investigate the impact of methionine restriction (MR) and cold exposure (CE) on liver and adipose tissues in mice. The authors demonstrate that responses to MR and CE are tissue-specific, while both MR and CE have a similar effect on beige adipose tissue. While these findings are somewhat descriptive, this work is considered **important**, as it provides a comprehensive resource for enhancing our understanding of these lifestyle interventions. The study is of high scientific quality, and the analyses are **convincing**.

*For correspondence:
pmr96@cornell.edu (PMMR);
janwilhelmkornfeld@bmb.sdu.dk (J-WK)

**Competing interest:** The authors declare that no competing interests exist.

## Abstract

About 1 billion people are living with obesity worldwide. GLP-1-based drugs have massively transformed care, but long-term consequences are unclear in part due to reductions in energy expenditure with ongoing use. Diet-induced thermogenesis (DIT) and cold exposure (CE) raise EE via brown adipose tissue (BAT) activation and beiging of white adipose tissue (WAT). Methionine restriction (MetR) is a candidate DIT stimulus, but its EE effect has not been benchmarked against CE, nor have their tissue-level interactions been defined. In a 2×2 design (Control vs. MetR; room temperature, RT: 22°C vs. CE: 4°C for 24 hr), we used male C57BL/6 N mice to benchmark MetR-induced thermogenesis against CE and mapped how diet and temperature interact across tissues. Bulk RNA-seq profiled liver, iBAT, iWAT, and eWAT. Differential expression was modeled with main effects and a diet × temperature interaction. KEGG GSEA was used to assess pathway-level enrichment. MetR increased EE at RT and shifted fuel use towards lipid oxidation, supporting MetR as a bona fide DIT stimulus. CE elevated EE across diets and blunted diet differences. Transcriptomic responses were tissue-specific: in liver, CE dominated gene induction while MetR and CE cooperatively repressed genes. The combination enriched glucagon/AMPK-linked and core metabolic pathways. In iBAT, CE dominated thermogenic and lipid-oxidation programs with minimal MetR contribution. In iWAT, MetR and CE acted largely additively with high concordance, enhancing fatty-acid degradation, PPAR signaling, thermogenesis, and TCA cycle pathways. In eWAT, robust co-dependent and synergistic differential expression emerged only with MetR+CE. MetR is a genuine DIT stimulus that remodels metabolism in a tissue-specific manner. Our study provides a tissue-resolved transcriptomic resource that benchmarks diet-induced (MetR) against cold-induced thermogenesis and maps their interactions across liver, iBAT, iWAT, and eWAT.

## Introduction

Obesity has become a global health crisis, with prevalence rates reaching alarming levels in many countries. The World Health Organization (WHO) estimates that over 890 million adults worldwide are obese (*Phelps et al., 2024*), leading to a dramatic increase in obesity-related diseases such as type 2 diabetes, cardiovascular disease, and certain cancers. As traditional weight-loss strategies often fail to provide lasting results, new therapeutic approaches are urgently needed. The increase in energy expenditure via brown adipose tissue (BAT) activation has emerged as a promising strategy for combating obesity. Unlike white adipose tissue (WAT), which stores excess energy, BAT is specialized in generating heat and driving energy expenditure via non-shivering thermogenesis, typically in response to cold exposure (CE). Pharmacologically or environmentally induced BAT activity results in improved metabolic health in mice and humans (*Schlein and Heeren, 2016*; *Yoneshiro et al., 2013*). Conversely, diet-induced thermogenesis (DIT) refers to the increase in energy expenditure associated with the digestion, absorption, and metabolism of food.

Recent studies have suggested that sulfur amino acid restriction, also referred to as methionine restriction (MetR), a dietary intervention that limits the intake of the amino acids methionine and cysteine, can enhance energy expenditure (EE) and promote metabolic health (*Fang et al., 2022*). Methionine and cysteine are amino acids involved in a plethora of metabolic reactions including protein synthesis, gene expression via methylation of DNA and histones, maintenance of DNA and RNA integrity via polyamine synthesis, redox balance via glutathione and $H_2S$ metabolism, and nucleotide biosynthesis via the folate cycle (*Fang et al., 2022*; *Forney et al., 2020*).

Studies have shown that MetR is sensed in the liver by the GCN2-PERK-ATF4-mediated integrated stress response (*Wanders et al., 2015*). Initially, it was thought that increases in circulating FGF21 drive increases in EE by activating UCP1-driven thermogenesis in brown adipose tissue via β-adrenergic (βr) signaling (*Fang et al., 2022*; *Babygirija and Lamming, 2021*). However, more recent studies also show that at least some of the metabolic benefits develop in βr-incompetent, FGF21-KO, and UCP1-KO animals, with or without increases in EE and independently of ambient temperature (*Forney et al., 2020*; *Wanders et al., 2015*; *Wanders et al., 2017*; *Lee et al., 2025*), highlighting the complexity of the physiological response to the dietary restriction of sulfur amino acids. Recently, it has been proposed that sulfur amino acid restriction results in metabolic inefficiency resulting in the excretion of various metabolites including β-hydroxybutyrate, pyruvate, citrate, alpha-ketoglutarate, carnosine, and others (*Varghese et al., 2025*).

The above-mentioned studies highlight an intriguing overlap between diet- and cold-induced thermogenesis, particularly in the context of energy expenditure regulation. Both stimuli rely on the activation of brown adipose tissue (BAT) and the 'beiging' of inguinal white adipose tissue (iWAT) to promote UCP1 expression and increase mitochondrial activity, and thereby boost calorie dissipation, potentially via similar signaling pathways, including the sympathetic nervous system and key metabolic regulators like FGF21 (*Spann et al., 2021*). However, whether diet- and cold-induced thermogenesis produces additive or synergistic effects on the activation of energy and systemic metabolism is unknown. Here, we systematically compare the physiological and transcriptional responses to MetR and CE across multiple metabolically active tissues. Using RNA sequencing, we dissect additive, synergistic, and antagonistic gene regulatory patterns and assess whether combining MetR with CE produces tissue-specific concordant or discordant novel transcriptional outcomes.

We demonstrate that MetR increased EE at RT and shifted fuel use toward lipid oxidation. CE elevated EE across diets and blunted diet differences. The transcriptional responses were tissue-specific at the gene and pathway level. Taken together, our results provide a unique and comprehensive gene regulatory framework to understand how dietary and environmental cues converge to shape tissue-specific gene expression programs and metabolic adaptation.

## Results

### Physiological and metabolic effects of methionine restriction (MetR) and cold exposure (CE)

To investigate the physiological impact of dietary sulfur amino acid content on EE under room temperature (RT, i.e., 22°C) and cold exposure (CE, 4°C for 24 hr), male C57BL/6 N mice were placed on one of three cysteine-depleted diets containing either 0.8% Methionine (Control; Ctrl), 0.12% Methionine

(Methionine-restricted; MetR) or 2% Methionine (Methionine-supplemented; MetS) for 6 and 5 days, respectively (*Figure 1A*, i.e. study 1). Following the diet switch from housing diet to aforementioned experimental diets, EE increased progressively in the MetR group under RT conditions, rising from approx. 0.45 kcal/hr to 0.6 kcal/hr. Conversely, EE did not change in the Ctrl and MetS groups (*Figure 1B*). On day 6, energy expenditure in the MetR group was significantly elevated compared to Ctrl and MetS groups, independent of starting body weight (*Figure 1C*, left). In contrast, CE elevated EE in all three diet groups to ~0.9 kcal/hr, thereby rescinding the differences between MetR and the other two diets at RT (*Figure 1A and C* right, *Figure 1—figure supplement 1A*). MetR-fed animals exhibited significantly greater body weight loss compared to Ctrl and MetS-fed animals over the total duration of experimental diet feeding (*Figure 1D*). As food intake and locomotor activity were similar across groups (*Figure 1—figure supplement 1B and D*), the greater weight loss in MetR-fed animals is likely attributable to increased energy expenditure under RT conditions.

Consistent with previous studies, MetR feeding also increased water intake (*Figure 1—figure supplement 1C*). Respiratory exchange ratios (RER), as an indicator for macronutrient fuel selection, progressively increased during light and dark phases in the Ctrl and MetS groups, indicating enhanced carbohydrate utilization, whereas MetR-fed animals demonstrated lower RER indices, indicating a preference for lipid oxidation. Upon CE, RER decreased in Ctrl and MetS-fed animals to levels comparable to those observed in MetR-fed animals (*Figure 1E*, *Figure 1—figure supplement 1F*). To further assess dietary and temperature interactions, we calculated daily averages of food and water intake, as well as locomotor activity and RER for each period. At RT, daily caloric intake was similar between groups, and CE led to a consistent increase in caloric intake in all groups (*Figure 1F*). Average daily water intake was unaffected by ambient temperature, whereas daily locomotor activity exhibited a downward trend in all three groups during cold exposure (*Figure 1F*, *Figure 1—figure supplement 1E*). Collectively, these data demonstrate that short-term MetR increases energy expenditure by ca. 20% at RT and, as a consequence, promotes a metabolic shift toward lipid oxidation, potentially due to a transcriptional induction of thermogenic processes and/or lipolysis in catabolic adipose tissue depots such as iWAT or BAT. However, CE as a major physiological stimulus to induce non-shivering thermogenesis (NST) was able to override this MetR-induced effect, prompting an additional ca. 60% increase in EE independent of experimental diet feeding and body weight, alongside a shift to lipid utilization.

To further investigate the potential interactions between MetR- and CE-induced thermogenesis on physiological, metabolic, and transcriptional parameters, a second experiment was conducted: Here, mice were divided into four groups and either fed Ctrl or MetR diets (see study 1) for seven days at 22°C, or additionally exposed to 4°C for 24 hr (Study 2). In the following, animals fed Ctrl or MetR diets for 7 days at 22°C are denominated as Ctrl_RT and MetR_RT, while animals fed these diets for 7 days before being subjected to 24 hr of cold are denominated as Ctrl_CE and MetR_CE (*Figure 1G*). All groups experienced significant body weight loss during the intervention (*Figure 1H*), presumably due to single-housing. Counterintuitively, Ctrl_CE did not display exacerbated weight loss compared to the Ctrl_RT group. By contrast, MetR_RT led to approximately 10% body weight loss, and the combined treatment (MetR_CE) resulted in further weight loss compared to MetR_RT alone (*Figure 1H*). The reductions in body weights coincided with organ-specific alterations in organ wet weights and organ/body weight ratios. Both Ctrl_CE and MetR_RT resulted in absolute (*Figure 1I*) and relative (*Figure 1—figure supplement 1G*) reductions in liver mass, and MetR_CE reduced absolute and relative liver mass further in an additive fashion. The stepwise reduction in relative liver mass indicates that liver atrophy partly accounted for the observed body weight loss, predominantly in MetR-exposed mice. In contrast, Ctrl_CE alone did not affect iWAT, gWAT, or iBAT mass. Intriguingly, under MetR_RT, all three adipose tissues showed (non-significant) upward trends in relative and absolute masses, which was exacerbated and partially significant under MetR_CE (*Figure 1I*, *Figure 1—figure supplement 1G*). Intriguingly, these data indicate that MetR_RT and Ctrl_CE synergize to promote increases in adipose tissue mass, particularly in gWAT and iBAT. To assess the associated systemic metabolic sequelae of MetR- and CE, we measured circulating metabolic parameters: Blood glucose levels did not change significantly but displayed a stepwise reduction between the groups, indicating additive effects between diet and ambient temperature (*Figure 1J*). In line with previous studies, serum triglycerides were reduced, albeit non-significantly, by Ctrl_CE and MetR_RT (*Grefhorst et al., 2018*; *Hasek et al., 2013*; *Bartelt et al., 2011*), but showed an even stronger reduction

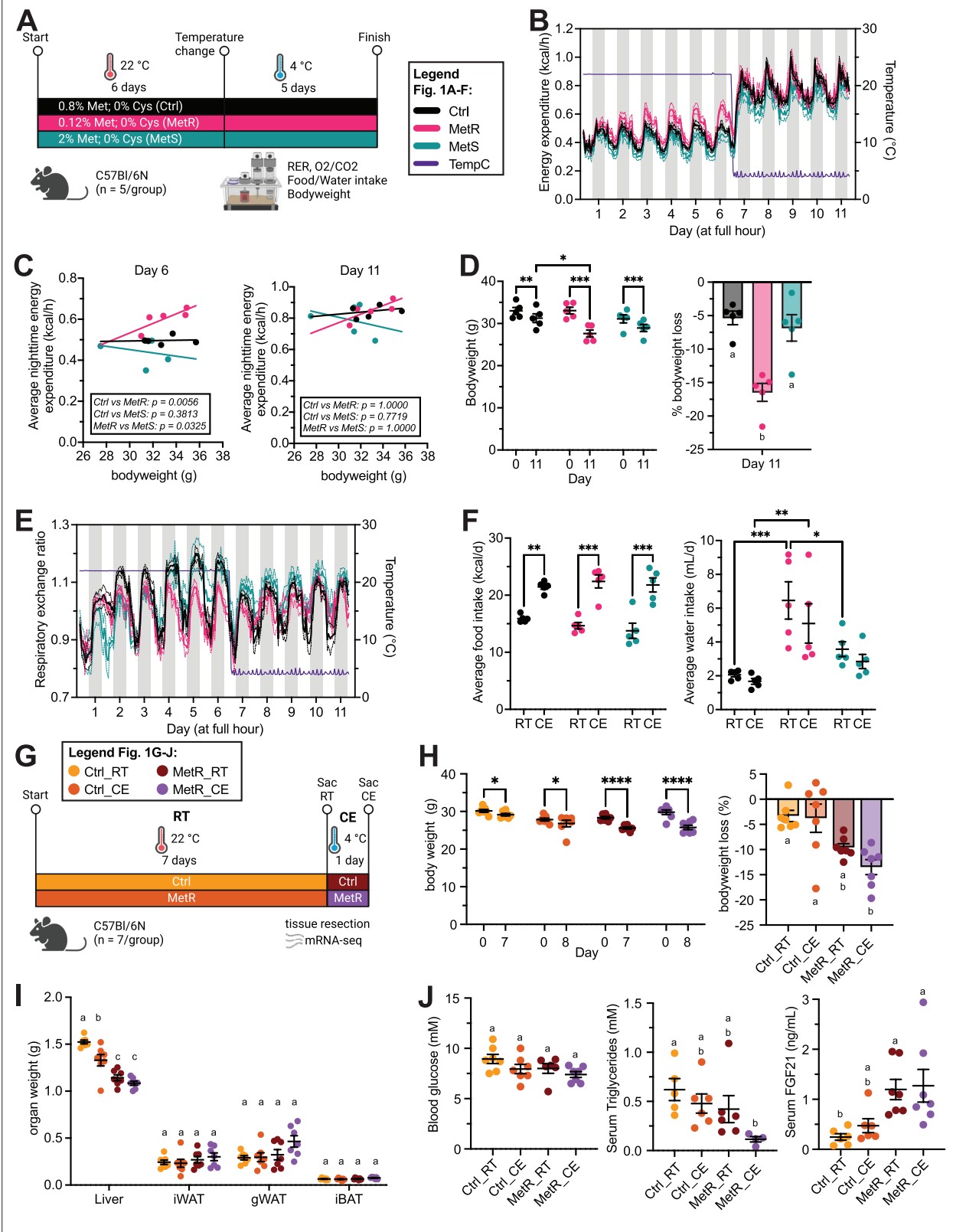

**Figure 1.** Physiological and metabolic effects of methionine restriction (MetR) and cold exposure (CE). (**A**) Schematic depicting 11-day experimental setup and dietary composition for mouse experiment 1 (n=5 animals/group). (**B**) Energy expenditure (EE) over the entire experiment duration. (**C**) ANCOVA analysis of average nighttime EE on days 6 and 11 over body weight. (**D**) Bodyweight and body weight loss (%). (**E**) Respiratory exchange ratio (RER) over the entire experiment duration. (**F**) Average daily food and water intake over all RT (22°C) or CE (4°C) experimental days. (**G**) Schematic

*Figure 1 continued on next page*

*Figure 1 continued*

depicting 8 day experimental setup for mouse experiment 2 (n=7 animals/group). Dietary compositions for Ctrl and MetR diets are the same as in mouse experiment 1. Groups are denominated as Ctrl_RT, Ctrl_CE, MetR_RT, and MetR_CE. (**H**) Bodyweight and body weight loss (%). (**I**) Absolute organ weights for Liver, inguinal WAT, gonadal WAT, interscapular BAT. Statistics were done within tissues. (**J**) Blood glucose, serum triglycerides, and FGF21 levels. EE was analyzed by ANCOVA using body weight as a covariate (via https://www.mmpc.org/); p-values were Bonferroni-corrected. Bodyweight, average food and water intake were analyzed via two-way ANOVA with *p < 0.05, **p < 0.01, ***p < 0.001, ****p < 0.0001. Body weight loss, serum parameters, and absolute organ weights were analyzed using one-way ANOVA with Tukey's post hoc test. Different letters indicate statistically significant differences (p < 0.05). Sample sizes for experiment 1 are 5 animals/group and 5–7 animals/group for Experiment 2. Error bars are SEM.

The online version of this article includes the following figure supplement(s) for figure 1:

**Figure supplement 1.** Physiological and metabolic effects of methionine restriction (MetR) and cold exposure (CE).

under MetR_CE, suggesting synergy (*Figure 1J*). By contrast, serum NEFA levels showed only minor non-significant reductions in both CE conditions and were unchanged by MetR_RT alone (*Figure 1J*), while serum β-hydroxybutyrate levels, a marker of hepatic fatty acid oxidation, were elevated by either CE or MetR feeding (*Figure 1—figure supplement 1H*). A major hormonal stimulus of EE in response to CE or MetR is mediated via 'thermogenic' hormones like FGF21 or IL-6. In line with the literature, Ctrl_CE and MetR_RT elevated circulating FGF21 levels even though the effect of cold was limited in this study. FGF21 levels did not further increase under MetR_CE, indicating that MetR maximally activates *Fgf21* transcription / FGF21 secretion to induce EE (*Figure 1J*). In contrast to literature, Ctrl_CE resulted in significant reductions in serum IL-6 levels in this study (*Figure 1—figure supplement 1H*; *Bal et al., 2017*). MetR_RT also reduced serum IL-6 levels, with no further alterations under MetR_CE (*Figure 1—figure supplement 1H*). Together, these results suggest that while cold exposure and MetR feeding individually trigger overlapping metabolic responses, their combination produces additive effects on weight loss, liver atrophy, and adipose size, alongside specific synergistic effects on triglyceride levels.

## The transcriptional responses to CE and MetR-induced thermogenesis are tissue-specific

To investigate the transcriptional basis of the adaptation to two independent, yet potentially synergistic, physiological stimulators of NST, we next performed bulk mRNA-seq on liver, iBAT, iWAT, and eWAT across all four groups. Principal Component Analysis (PCA), as expected, revealed that the samples clustered according to their tissue of origin (*Figure 2—figure supplement 1A*). We next performed tissue-intrinsic PCA analyses that demonstrated highly tissue-specific responses (*Figure 2A*). *Phelps et al., 2024* In the Liver, all four groups showed distinct transcriptional responses in PC1 and PC2, whereas in *Schlein and Heeren, 2016* iBAT the groups clustered mostly by temperature (PC1), indicating that the transcriptional effects of MetR feeding are mild compared to CE. By contrast, in both white adipose depots, the four groups showed significant overlap. Although, in *Yoneshiro et al., 2013* iWAT, the clusters showed additive effects of MetR and CE along PC1, explaining 60% of the variation, while in *Fang et al., 2022* eWAT, the clusters overlapped in PC1 and PC2. Differential gene expression analysis (cutoff: adj. pvalue <0.05, |FC| ≥ 1.5) showed that Ctrl_CE elicited stronger responses than MetR_RT in all tissues as exemplified by the higher number of differentially expressed genes (DEGs) without clear trends towards global transcriptional induction or repression. Of note, MetR was coupled to a higher number of repressed genes, potentially linked to the increased need for cellular energy conservation during amino acid restriction. The combination of both stimuli elicited the strongest transcriptional responses in iWAT, and apart from the interaction term, the transcriptional response in eWAT appeared limited (*Figure 2*, *Figure 2—figure supplement 1*).

To understand to what degree the transcriptional responses are tissue/depot-specific or broadly applicable to all investigated organs, we analyzed the intersections between CE and MetR in all four tissues (*Figure 2C–F*, *Figure 2—figure supplement 1C and D*). Under Ctrl_CE and MetR_CE, more than 50% of all DEGs were regulated in a tissue-specific manner in three out of the four tissues, with the most marked transcriptional effects seen in iWAT (*Figure 2C–E*). The degree of tissue-specific regulation was even higher in MetR_RT and in the interaction term (*Figure 2D and F*). Cumulatively, these data suggest that the majority of transcriptional effects of CE and dietary MetR are tissue-specific and display a varying degree of additive and synergistic effects.

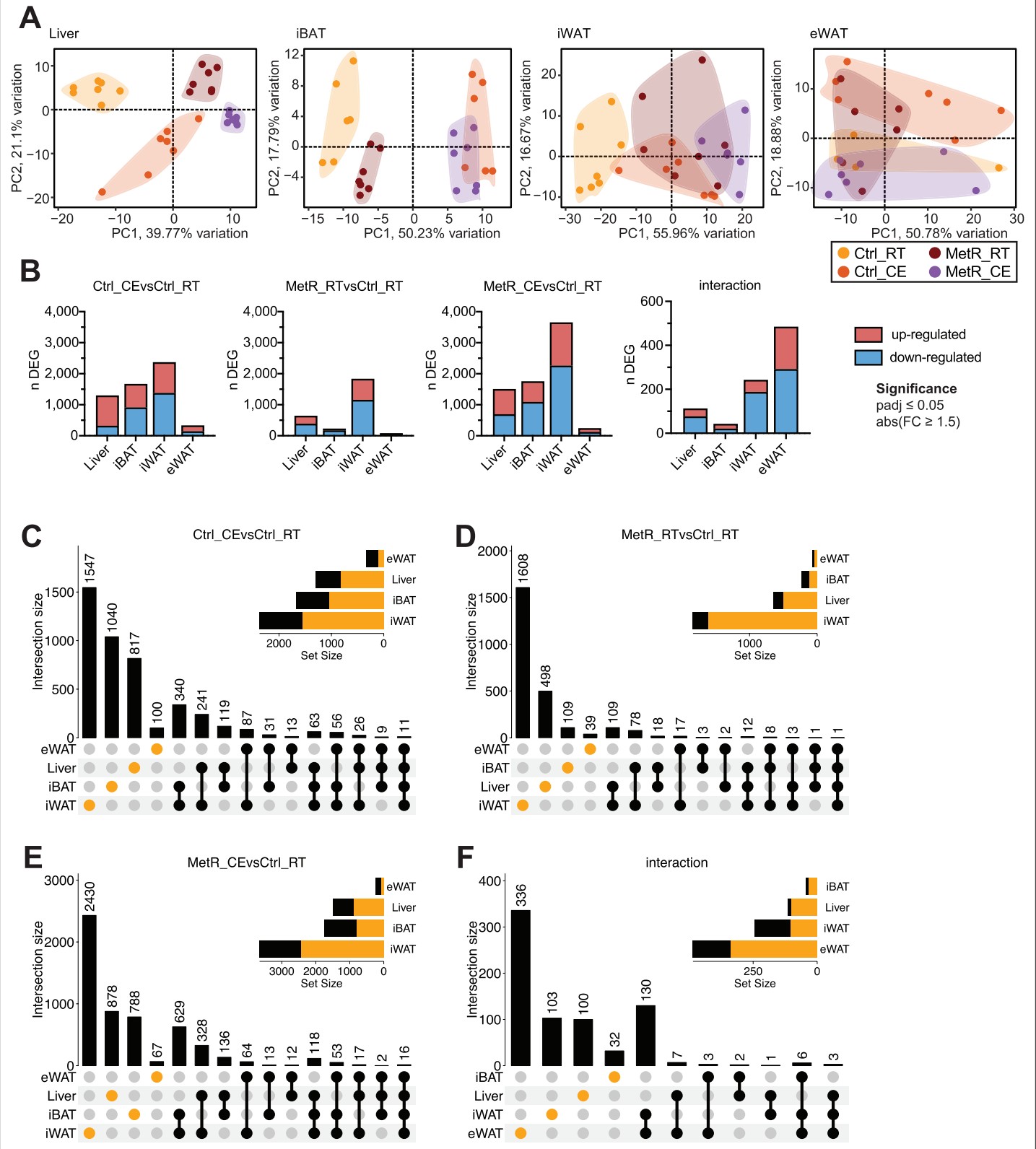

**Figure 2.** The transcriptional responses to cold exposure (CE) and MetR-induced thermogenesis are tissue-specific. (**A**) Tissue-specific PCA plots for liver, iBAT, iWAT, and eWAT. (**B**) Number of differentially expressed genes (DEGs), split into induced (red) and repressed (blue) genes, per contrast (adjusted p-value <0.05, |FC|>1.5). (**C–F**) UpSet plots showing overlap of DEGs across tissues for each contrast: (**C**) Ctrl_CE vs Ctrl_RT, (**D**) MetR_RT vs Ctrl_RT, (**E**) MetR_CE vs Ctrl_RT, and (**F**) Interaction. Set size bars indicate the total number of DEGs per tissue; intersection bars indicate the number

*Figure 2 continued on next page*

*Figure 2 continued*

of shared DEGs between tissues. Dots and bars in orange represent tissue-specific DEGs. In the order of Ctrl_RT, Ctrl_CE, MetR_RT, and MetR_CE, the following number of samples were included in the transcriptomic analysis: 7, 7, 7, and 7 (Liver); 7, 6, 7, and 7 (iBAT); 7, 7, 7, and 7 (iWAT); and 4, 7, 6, and 7 (eWAT; see Materials and methods).

The online version of this article includes the following figure supplement(s) for figure 2:

**Figure supplement 1.** The transcriptional responses to cold exposure (CE) and MetR-induced thermogenesis are tissue-specific.

To better scrutinize the individual transcriptional effects for each tissue, and to associate the observed gene-regulatory effects with cellular and metabolic processes, we next performed binary, combinatorial analyses of differential gene expression in liver (*Figure 3*), iBAT (*Figure 4*), iWAT (*Figure 5*), and eWAT (*Figure 6*) using similar in silico approaches.

## CE drives gene induction while MetR and CE cooperatively repress genes in the liver

In the liver (*Figure 3*), Volcano plot analysis revealed substantial gene regulation induced by Ctrl_CE (977 genes upregulated, 322 genes downregulated) and MetR_RT (257 genes upregulated, 387 genes downregulated), with even more genes regulated when combined under MetR_CE (815 genes upregulated, 692 genes downregulated). While Ctrl_CE appears to be a stronger stimulus based on the number of DEGs, MetR_RT appears to elicit stronger transcriptional effects based on $\log_2$FC (*Figure 3A*). Based on the number of DEGs, MetR_RT and Ctrl_CE, when combined under MetR_CE, appear to repress gene expression in an independent, additive manner and induce genes in a dependent or antagonistic manner (*Figure 3A*). Indeed, only a very limited number of genes were significant in diet × temperature interaction term (36 genes upregulated, 77 genes downregulated; *Figure 3A*) and the transcriptional effects of CE under MetR (MetR_CE vs MetR_RT) and MetR feeding under CE (MetR_CE vs Ctrl_CE) were lower compared to their respective counterparts (*Figure 3—figure supplement 1A*). To gain further insights into the degree of overlap between Ctrl_CE and MetR_RT, we performed Upset plot analysis. The biggest set of shared genes was found between Ctrl_CE and MetR_CE (541 DEGs), highlighting that Ctrl_CE is a major contributor to the transcriptional effects seen in MetR_CE (*Figure 3B*). While the direct contribution of MetR_RT alone was lower (251 DEGs), this analysis also indicated a high degree of co-dependence on MetR_RT in the regulation of 508 DEGs in the MetR_CE condition, as well as the exclusion of 494 Ctrl_CE DEGs from the combined exposure (*Figure 3B*). Contrasting all MetR_CE DEGs between Ctrl_CE and MetR_RT ($R^2$=0.34), as well as contrasting all MetR_RT and Ctrl_CE DEGs ($R^2$=0.26), revealed relatively poor correlation between the respective $\log_2$FC values and suggests that the overall influence of Cold is stronger (*Figure 3C*, *Figure 3—figure supplement 1B*). Next, we analyzed the transcriptional interaction between Ctrl_CE and MetR_RT (*Figure 3D*, *Figure 3—figure supplement 1C*, see methods) under consideration of significance, direction of regulation, and interaction significance based on the method proposed by *Pantaleón García et al., 2022*. This analysis indicated that more than 50% of down-regulated genes were due to co-dependent additive regulation between Ctrl_CE and MetR_RT (brown) and MetR_RT alone (green). By contrast, more than 50% of the up-regulated genes were significant in the combined exposure due to Ctrl_CE alone (*Figure 3D*). Altogether, these data confirm Ctrl_CE drives the induction of genes under MetR_CE, while Ctrl_CE and MetR_RT cooperate to repress genes under MetR_CE. On the other hand, MetR_RT prevented a significant number of Ctrl_CE DEGs (n=547 genes) from regulation under MetR_CE in an additive (subtractive) fashion (*Figure 3—figure supplement 1D*). Finally, to understand which pathways are differentially regulated, we performed Gene set enrichment analysis (GSEA) under consideration of $\log_2$FC values (*Subramanian et al., 2005*). Among the 20 most enriched pathways under MetR_CE, MetR drove the enrichment for metabolic pathways (Biosynthesis of amino acids, Glutathione or P450 cytochrome (Drug) metabolism) despite the effects of Ctrl_CE, while CE drove the enrichment in Steroid biosynthesis and cGMP-PKG signaling pathways, despite the effects of MetR_RT (*Figure 3F*). These results are in line with previously published datasets (*Zhang et al., 2022*; *Ghosh et al., 2017*). This analysis also revealed a number of gene sets where MetR_CE showed the highest enrichment, such as Glucagon and AMPK signaling, TCA cycle, One carbon metabolism, and amino acid biosynthesis. This indicates that MetR and CE cooperate in the regulation of gene sets related to energy provisioning and glucose metabolism in the liver

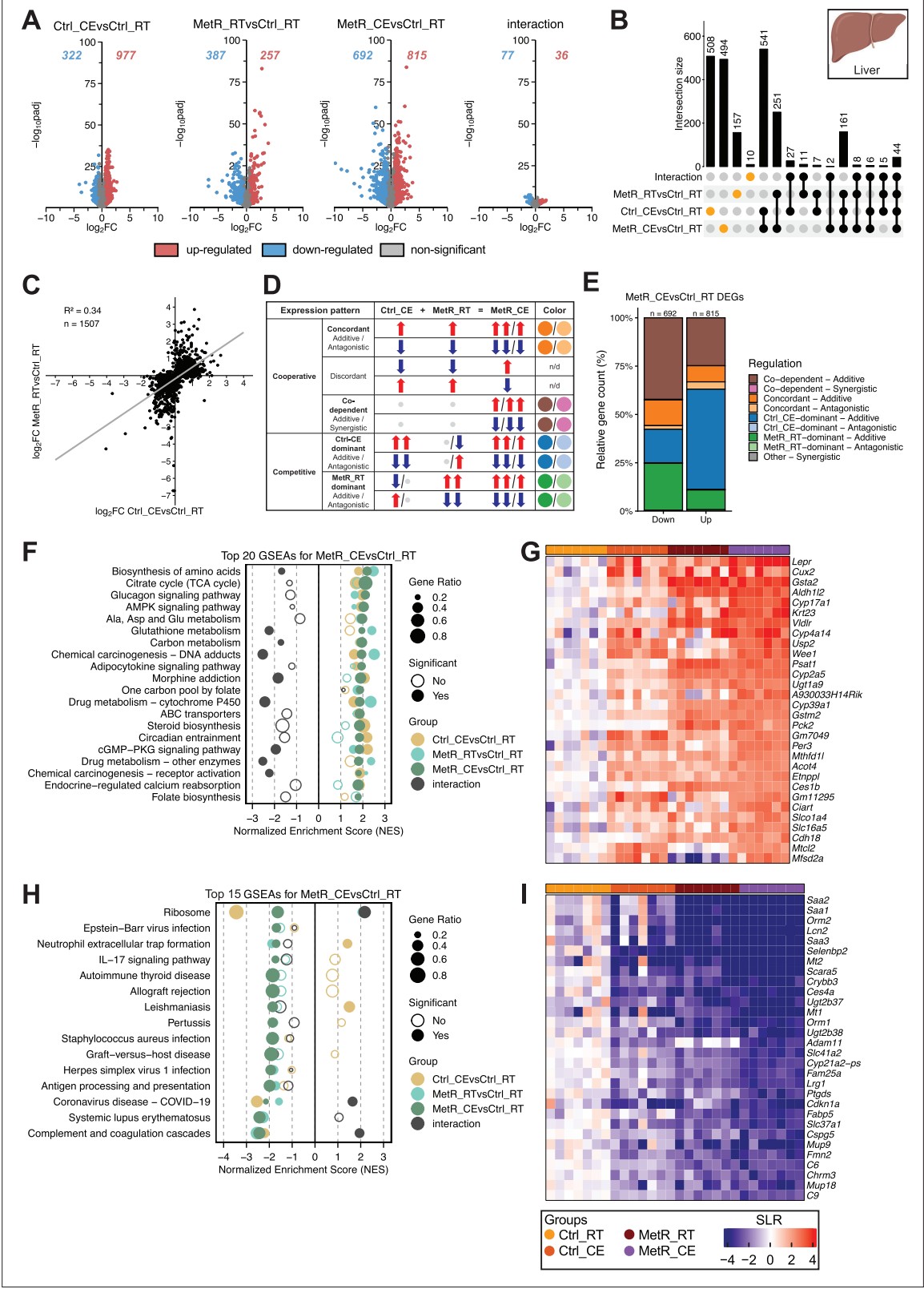

**Figure 3.** Cold exposure (CE) drives gene induction while methionine restriction (MetR) and CE cooperatively repress genes in the liver. (**A**) Volcano plots showing DEGs for each contrast (Ctrl_CE vs Ctrl_RT, MetR_RT vs Ctrl_RT, MetR_CE vs Ctrl_RT, and the diet × temperature interaction). Numbers indicate significantly up- and downregulated genes (adjusted p-value <0.05, |FC|>1.5). (**B**) Upset plot showing the overlap of DEGs across contrasts. Intersection bars indicate the number of shared DEGs between contrasts. Dots in orange represent contrast-specific DEGs. (**C**) Scatter plot of log₂FC

*Figure 3 continued*

values in Ctrl_CE vs Ctrl_RT and MetR_RT vs Ctrl_RT for DEGs identified in MetR_CE. n denominates DEGs shown. (**D**) Schematic highlighting gene expression profiles. Arrows indicate significant regulation for induced (up; red) or repressed (down; blue) genes. Non-significant regulation is depicted as gray dots (see Materials and methods). (**E**) Classification of MetR_CE DEGs based on their mode of regulation mode. (**F**) GSEA of the top 20 positively enriched pathways in MetR_CE vs Ctrl_RT. Dot size represents gene ratio, color denotes contrast, and significance is indicated by filled circles (adjusted p-value <0.05). (**G**) Heatmap of Top 30 induced genes under MetR_CE. (**H**) GSEA for the top 14 negatively enriched pathways in the MetR_CE vs Ctrl_RT comparison. Dot size reflects gene ratio, colors indicate contrast group, and significance is shown by filled circles (adjusted p-value <0.05). (**I**) Heatmap of Top 30 repressed genes under MetR_CE.

The online version of this article includes the following figure supplement(s) for figure 3:

**Figure supplement 1.** Cold exposure (CE) drives gene induction while methionine restriction (MetR) and CE cooperatively repress genes in the liver.

(*Figure 3F*). The top 30 heatmap of induced genes under MetR_CE reflects these additive effects of Ctrl_CE and MetR_RT on the level of pathway enrichment and features genes such as *Pck2* and *Lepr* (Glucagon signaling), *Pck2* (TCA cycle), *Mthfd1l* and *Aldh1l2* (TCA cycle and folate metabolism), and *Psat1*, *Etnppl,* and *Acot4* (amino acid biosynthesis; *Figure 3G*). The only non-immune-related pathway among the 14 most negatively enriched pathways was the Ribosome gene set, where Cold and MetR feeding had opposing effects (*Figure 3H*). In accordance with the contrasting enrichment by MetR_RT and Ctrl_CE alone, the top 30 heatmap of MetR_CE repressed genes only features immune-related genes including *Saa1*, *Saa2*, *Saa3* and *Orm1* and *Orm2* (acute-phase response), *Lcn2* and *Lrg1* (innate immunity; *Figure 3I*). Furthermore, additional heatmap analyses of the top 30 up- and down-regulated genes for Ctrl_CE and MetR_RT further highlight the distinct and cooperative transcriptional effects between both stimuli (*Figure 3—figure supplement 1E–J*).

## CE dominates gene induction in iBAT with limited contribution by MetR feeding

In iBAT (*Figure 4*), Volcano plot analysis revealed that Ctrl_CE elicited a much stronger transcriptional response than MetR_RT (757 genes upregulated, 912 genes downregulated, versus 57 genes upregulated and 173 downregulated, respectively). The transcriptional response to MetR_CE was only marginally different from Ctrl_CE (664 genes upregulated, 1091 genes downregulated), suggesting that the majority of the transcriptional response is driven by CE (*Figure 4A*). Only a very small number of genes were identified in the diet ×temperature interaction term (22 upregulated, 21 downregulated), indicating that the combination of CE and MetR does not result in non-additive transcriptional effects in iBAT (*Figure 4A*). In line with this, CE on top of MetR (MetR_CE vs MetR_RT) still resulted in substantial gene regulation (*Figure 4—figure supplement 1A*). Upset plot analysis confirmed that the majority of DEGs in the combined MetR_CE condition overlapped with Ctrl_CE (1009 genes), while only a minor subset was shared with MetR_RT (85 genes; *Figure 4B*). Comparison of log$_2$FC between Ctrl_CE and MetR_RT for all MetR_CE DEGs showed a weak correlation (R$^2$=0.24), supporting the notion that the transcriptional responses to MetR_CE are indeed driven by CE (*Figure 4C*). This weak correlation was confirmed when considering all MetR_RT or Ctrl_CE DEGs (*Figure 4—figure supplement 1B*). The classification of gene regulation confirmed that >50% of down- and up-regulated genes were driven by Ctrl_CE (blue, *Figure 4D*). Altogether, these data indicate that Ctrl_CE primarily drives gene regulation in iBAT, while the transcriptional effects of MetR_RT are negligible. GSEA analysis confirmed the positive enrichment of fatty acid and thermogenic pathways by Ctrl_CE, as previously published (*Ghosh et al., 2017*; *Hao et al., 2015*). The combination of both stimuli specifically benefited the positive enrichment of metabolic pathways such as fatty acid elongation, biosynthesis of unsaturated fatty acids, and PPAR signaling, while Ctrl_CE drove the enrichment in ER and glycerolipid-related gene sets. In line with the limited transcriptional response of MetR_RT on gene level, NES scores for MetR_RT were mostly lower compared to Ctrl_CE and/or non-significant, except for Steroid biosynthesis (*Figure 4E*). In line, the heatmap of the top 30 upregulated genes under MetR_CE underscores this enrichment by featuring important factors such as *Ppargc1a*, *Fgf21*, *Gdf15*, *Gpr3*, and *Bmp8b* (PPAR signaling and thermogenesis), *Elovl3* and *Scd3* (fatty acid elongation), and *Mthfd2* (folate metabolism; *Figure 4F*). Many of these genes are also featured in the top 30 heatmap for Ctrl_CE regulated genes (*Figure 4—figure supplement 1D*). By contrast, negatively enriched gene sets under MetR_CE were largely immune- and extracellular matrix–related, including cytokine-, ECM- and neuroactive ligand-receptor interactions and cell adhesion gene sets (*Col1a1, Col1a2, Ngfr,*

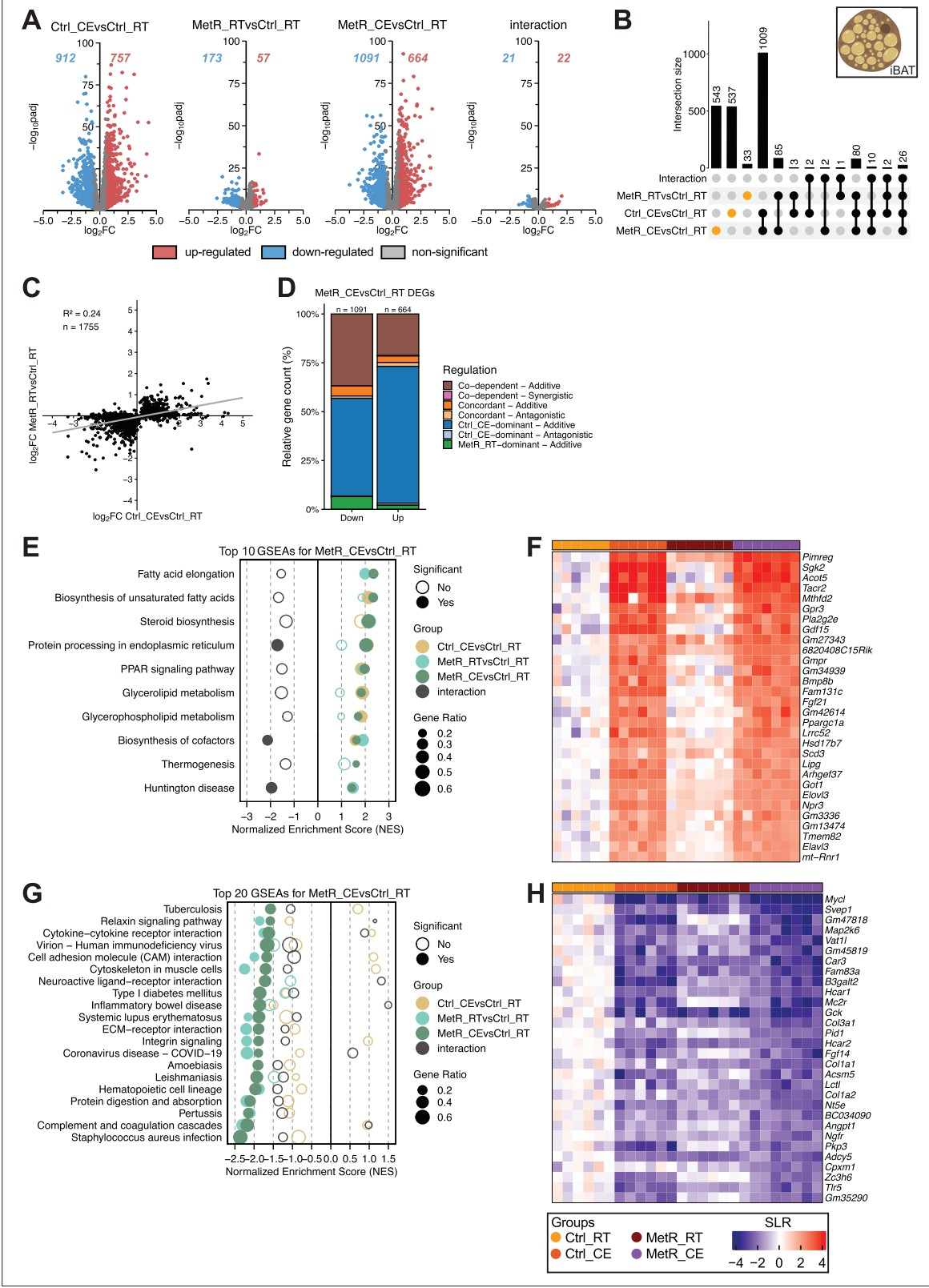

**Figure 4.** CE dominates gene induction in iBAT with limited contribution by MetR feeding. (**A**) Volcano plots showing DEGs for each contrast (Ctrl_CE vs Ctrl_RT, MetR_RT vs Ctrl_RT, MetR_CE vs Ctrl_RT, and the diet × temperature interaction). Numbers indicate significantly up- and downregulated genes (adjusted p-value <0.05, |FC|>1.5). (**B**) Upset plot showing the overlap of DEGs across contrasts. Intersection bars indicate the number of shared DEGs between contrasts. Dots in orange represent contrast-specific DEGs. (**C**) Scatter plot of log₂FC values in Ctrl_CE vs Ctrl_RT and MetR_RT vs

*Figure 4 continued on next page*

*Figure 4 continued*

Ctrl_RT for DEGs identified in MetR_CE. n denominates DEGs shown. (**D**) Classification of MetR_CE DEGs based on their mode of regulation. (**E**) GSEA of the top 10 positively enriched pathways in MetR_CE vs Ctrl_RT. Dot size represents gene ratio, color denotes contrast, and significance is indicated by filled circles (adjusted p-value <0.05). (**F**) Heatmap of Top 30 induced genes under MetR_CE. (**G**) GSEA for the top 20 negatively enriched pathways in the MetR_CE vs Ctrl_RT comparison. Dot size reflects gene ratio, colors indicate contrast group, and significance is shown by filled circles (adjusted p-value <0.05). (**H**) Heatmap of Top 30 repressed genes under MetR_CE.

The online version of this article includes the following figure supplement(s) for figure 4:

**Figure supplement 1.** CE dominates gene induction in iBAT with limited contribution by MetR feeding.

*Hcar2, Adcy5*). Interestingly, the negative enrichment of these pathways was dominated by MetR_RT (*Figure 4G and H*). Altogether, these analyses support the notion that the transcriptional effects of MetR_RT are limited and contribute little to the combined effect of MetR_CE, which is dominated by cold (*Figure 4—figure supplement 1D–G*).

## Additive and synergistic gene regulation by MetR and CE in iWAT

In iWAT (*Figure 5*), Volcano plots revealed that both Ctrl_CE (992 upregulated, 1379 genes downregulated) and MetR_RT (680 upregulated, 1156 downregulated) elicited marked transcriptional changes: The combination of both stimuli (MetR_CE) resulted in an even greater number of DEGs compared to either stimuli alone (1394 upregulated, 2261 downregulated), indicating either additive or synergistic effects (*Figure 5A*). Only a limited number of genes were identified in the diet × temperature interaction (188 down, 55 up), suggesting few non-additive responses (*Figure 5A*). Upset plots suggest cooperativity between MetR_RT and Ctrl_CE, with 1234 DEGs uniquely regulated when both stimuli were combined, but also support true additive or synergistic regulation with 1210 genes being significantly regulated in all three conditions (*Figure 5B*). Correlation of log$_2$FC between Ctrl_CE and MetR_RT for all MetR_CE showed a strong positive correlation (R$^2$=0.81), suggesting that the impact of CE and MetR feeding on gene regulation is similar in iWAT (*Figure 5*, *Figure 5—figure supplement 1*). Regulatory classification of MetR_CE DEGs revealed that 55% of up- and 75% of down-regulated genes were predominantly regulated in an additive manner, where Ctrl_CE and MetR_RT cooperated to enhance transcriptional responses. This included genes that became significant when both stimuli were combined (co-dependent regulation) as well as genes that were already significantly regulated under each condition alone but showed non-synergistic regulation under the combined treatment (concordant – additive; *Figure 5D*). GSEA analysis revealed in pathways consistent with expected CE- or MetR-induced responses, such as fatty acid degradation, PPAR signaling, thermogenesis, and the TCA cycle (*Figure 5E*; *Ghosh et al., 2017*; *Hao et al., 2015*). Notably, nearly all of these pathways showed significant negative enrichment for the interaction term, indicating that the transcriptional response to MetR_CE is less than additive. This suggests that while both Ctrl_CE and MetR_RT independently activate key metabolic pathways, their combined effect is attenuated on the gene set level, possibly due to overlapping mechanisms or transcriptional feedback limiting further activation. Heatmap analysis of the top 30 induced genes matches GSEA results and features prominent PPAR and thermogenesis-related genes including *Dio2, Ucp1, Fgf21, Elovl3, Cpt1b*, and *Ppargc1a*. The heatmap analysis also confirms the largely additive gene regulatory effects of Ctrl_CE and MetR_RT on gene level (*Figure 5F*, *Figure 5—figure supplement 1D and E*). Similar to the responses in iBAT, iWAT also showed negative enrichment in predominantly immune-related pathways, such as cytokine-cytokine receptor interactions (*Ccl2, Ccl7, Ccl8, Ccl12, Ccr5, Il1rl1*; *Figure 5G and H*).

## Limited yet codependent and antagonistic gene regulation by MetR and CE in eWAT

In eWAT (*Figure 6*), Volcano plot analysis revealed that both CE and MetR feeding alone did not have substantial effects on transcription, as less than 189 DEGs were up- or down-regulated in either condition (*Figure 6A*). This combined effect under MetR_CE also featured merely 122 up- or down-regulated genes, while the interaction term identified 193 up- and 292 down-regulated DEGs, the highest among all tissues, suggesting a potential synergistic mode of gene regulation under MetR_CE in this tissue (*Figure 6A*). Upset plot analysis confirmed that most MetR_CE DEGs (158) are unique to the combination of CE and MetR, and that Ctrl_CE (31 DEGs) or MetR_RT (24 DEGs) alone contributed

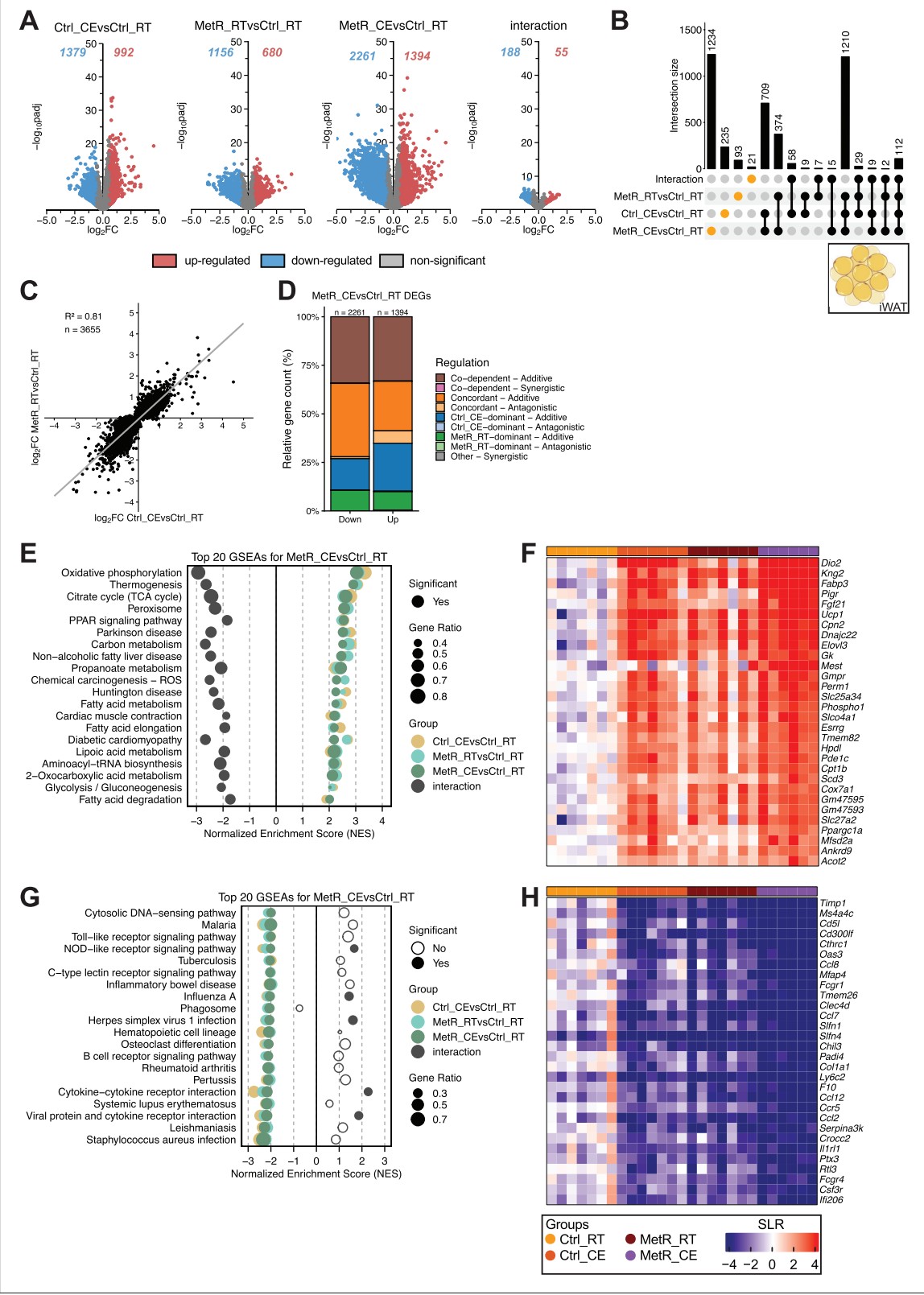

**Figure 5.** Additive and synergistic gene regulation by MetR and CE in iWAT. (**A**) Volcano plots showing DEGs for each contrast (Ctrl_CE vs Ctrl_RT, MetR_RT vs Ctrl_RT, MetR_CE vs Ctrl_RT, and the diet × temperature interaction). Numbers indicate significantly up- and downregulated genes (adjusted p-value <0.05, |FC|>1.5). (**B**) Upset plot showing the overlap of DEGs across contrasts. Intersection bars indicate the number of shared DEGs between contrasts. Dots in orange represent contrast-specific DEGs. (**C**) Scatter plot of log₂FC values in Ctrl_CE vs Ctrl_RT and MetR_RT vs Ctrl_RT for

*Figure 5 continued on next page*

Figure 5 continued

DEGs identified in MetR_CE. n denominates DEGs shown. (**D**) Classification of MetR_CE DEGs based on their mode of regulation. (**E**) GSEA of the top 20 positively enriched pathways in MetR_CE vs Ctrl_RT. Dot size represents gene ratio, color denotes contrast, and significance is indicated by filled circles (adjusted p-value <0.05). (**F**) Heatmap of Top 30 induced genes under MetR_CE. (**G**) GSEA for the top 20 negatively enriched pathways in the MetR_CE vs Ctrl_RT comparison. Dot size reflects gene ratio, colors indicate contrast group, and significance is shown by filled circles (adjusted p-value <0.05). (**H**) Heatmap of Top 30 repressed genes under MetR_CE.

The online version of this article includes the following figure supplement(s) for figure 5:

**Figure supplement 1.** Additive and synergistic gene regulation by MetR and CE in iWAT.

little to the joined MetR_CE effect (*Figure 6B*). Scatter plot analysis of $\log_2$FC values between Ctrl_CE and MetR_RT for all MetR_CE DEGs revealed a relatively high correlation ($R^2$=0.54) and were predominantly categorized as Co-dependent–additive (brown; *Figure 6C and D*). Conversely, all DEGs from either Ctrl_CE or MetR_RT correlated less with each other ($R^2$=0.29) and were found to interact in an antagonistic fashion (*Figure 6—figure supplement 1B*, C). This suggests a complex gene-regulatory relationship between MetR feeding and CE in eWAT. On the one hand, MetR_RT and Ctrl_CE additively regulate genes that only become significant under MetR_CE; on the other hand, DEGs unique to MetR_RT or Ctrl_CE alone are excluded from the combined MetR_CE response in an antagonistic fashion. GSEA analysis revealed that significant positive enrichment in immune and signaling pathways such as Rap1 signaling, MAPK signaling, and cGMP-PKG signaling (*Maff*, *Ptchd4*, and *Arhgef37*) under the combined treatment of MetR_CE (*Figure 6E and F*). Here, interaction between MetR_RT and Ctrl_CE enrichment scores appeared additive in nature (non-significant enrichment for interaction DEGs). Intriguingly, negatively enriched pathways under MetR_CE were largely associated with fatty acid metabolism and oxidative phosphorylation (*Figure 6G*). While MetR_RT alone again did not result in significant enrichment here, Ctrl_CE often resulted in opposing (significant or non-significant) enrichment for the mentioned pathways. These enrichments match with repressed genes that are regulated in an additive manner between MetR_RT and Ctrl_CE (*Cyp2e1*, *Cyp4f17*, *Ehhadh*, *Slc16a1*, *Serpina3b*; *Figure 6H*) and genes that are regulated in an antagonistic manner (*Acly*, *Elovl6*, *Angptl8*, *Acss2*, *Me1*, *Fasn*, *Scd2*, *Acaca*, *Slc25a1*; *Figure 6—figure supplement 1F*, I). Overall, these results highlight a complex relationship between both thermogenic stimuli in eWAT comprising both additive but also antagonistic regulation at the gene and pathway level.

## Discussion

In this work, we compared the systemic effects of two environmental stimuli of energy expenditure, dietary MetR and CE, in a 2x2 design, in order to assess whether both stimuli produce additive or synergistic systemic energy metabolism and transcriptional adaptations in key metabolic tissues. Our results indicate that although dietary MetR increased EE at RT and shifted substrate utilization towards lipid oxidation, CE constituted a stronger stimulus for EE and masked diet-dependent differences. Combining CE and MetR resulted in mild reductions in body weight and glucose levels and additive reductions in circulating triglyceride levels and liver mass. Notwithstanding the threshold effects seen in the activation of FGF21 and β-hydroxybutyrate (βOHB), markers for energy homeostasis and hepatic fatty acid oxidation, respectively, these results suggest that there is potential in combining both stimuli for the correction of hyperglycemia, hypertriglyceridemia, and excess body weight in disease settings.

To gain further insights into how MetR and CE converge on the transcriptional level, we performed RNA-seq on liver, iBAT, iWAT, and eWAT. We observed a high degree of depot-specificity in the transcriptional responses to MetR, CE, and the combination, both at the gene and pathway level. In the liver, CE dominated gene induction while MetR and CE cooperatively repressed genes. The combination of both factors regulated pathways involved in energy and glucose handling (glucagon signaling, TCA cycle, AMPK signaling, amino acid and carbon metabolism), potentially explaining the stepwise reduction in circulating glucose and triglyceride levels. In iBAT, CE dominated the transcriptional response, with limited contribution from MetR, and resulted in the enrichment of thermogenic and lipid-oxidation programs (thermogenesis, fatty acid metabolism). In iWAT, MetR and CE acted largely additively with high concordance, enhancing fatty-acid degradation, PPAR signaling, thermogenesis, and TCA cycle pathways. The transcriptional responses in eWAT overall were limited,

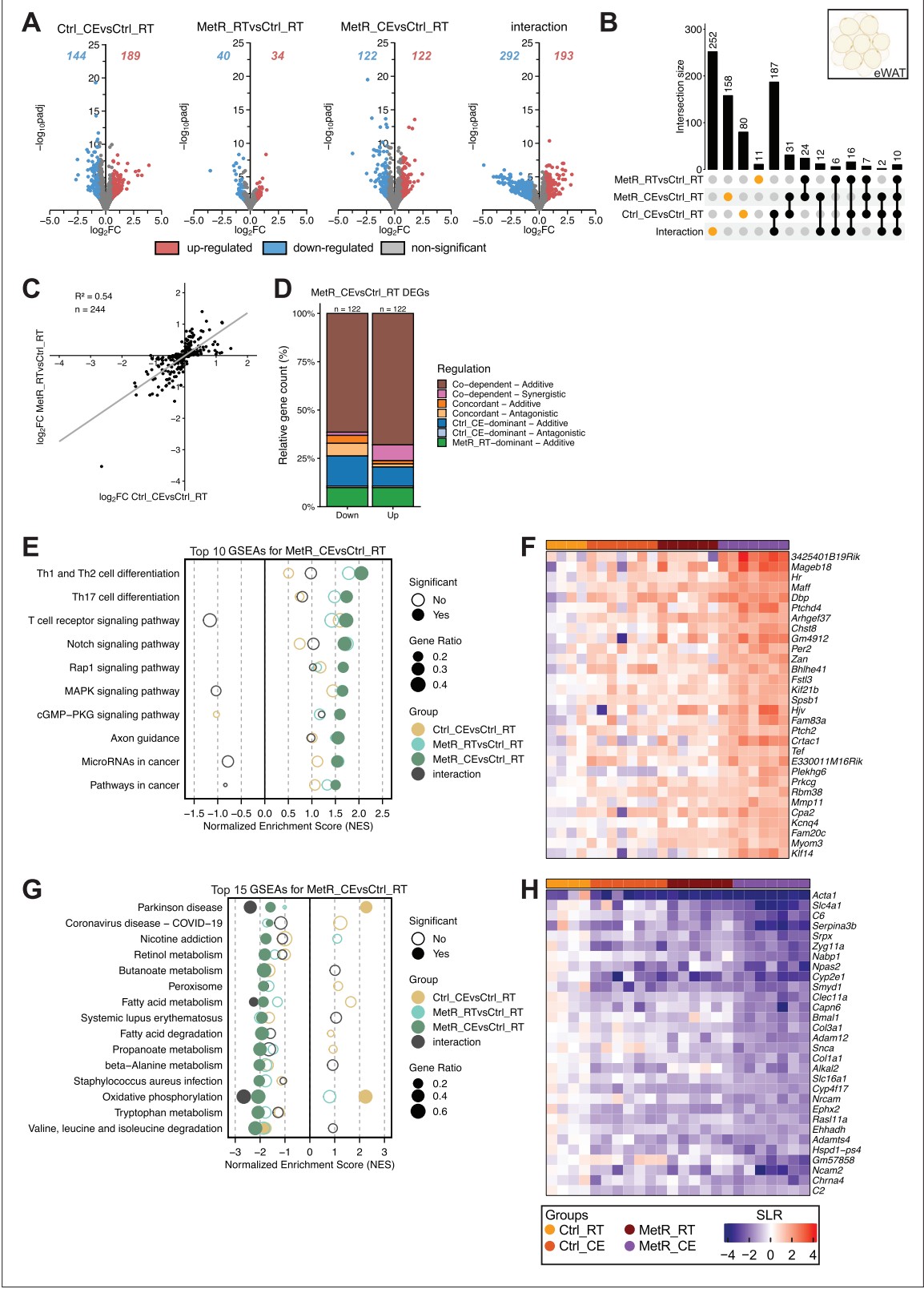

**Figure 6.** Limited yet codependent and antagonistic gene regulation by MetR and CE in eWAT. (**A**) Volcano plots showing DEGs for each contrast (Ctrl_CE vs Ctrl_RT, MetR_RT vs Ctrl_RT, MetR_CE vs Ctrl_RT, and the diet × temperature interaction). Numbers indicate significantly up- and downregulated genes (adjusted p-value <0.05, |FC|>1.5). (**B**) Upset plot showing the overlap of DEGs across contrasts. Intersection bars indicate the number of shared DEGs between contrasts. Dots in orange represent contrast-specific DEGs. (**C**) Scatter plot of log₂FC values in Ctrl_CE vs Ctrl_RT and MetR_RT vs

*Figure 6 continued on next page*

*Figure 6 continued*

Ctrl_RT for DEGs identified in MetR_CE. n denominates DEGs shown. (**D**) Classification of MetR_CE DEGs based on their mode of regulation. (**E**) GSEA of the top 10 positively enriched pathways in MetR_CE vs Ctrl_RT. Dot size represents gene ratio, color denotes contrast, and significance is indicated by filled circles (adjusted p-value <0.05). (**F**) Heatmap of Top 30 induced genes under MetR_CE. (**G**) GSEA for the top 20 negatively enriched pathways in the MetR_CE vs Ctrl_RT comparison. Dot size reflects gene ratio, colors indicate contrast group, and significance is shown by filled circles (adjusted p-value <0.05). (**H**) Heatmap of Top 30 repressed genes under MetR_CE.

The online version of this article includes the following figure supplement(s) for figure 6:

**Figure supplement 1.** Limited yet codependent and antagonistic gene regulation by MetR and CE in eWAT.

producing some additional, yet mostly antagonistic effects. Whether the downregulation of fatty acid metabolism pathways is linked to the relative increase of eWAT mass under MetR_CE requires further investigation.

An intriguing finding of potential translational value was the cold-like transcriptional response elicited by MetR feeding in iWAT. Amidst the emerging appreciation that the amount and metabolic activity of 'classical' human brown adipocytes might not contribute to overall EE and metabolic regulation in humans (**Becher et al., 2021**), other fat depots such as 'beige' subcutaneous adipose tissue (scWAT, i.e. the functional human equivalent to murine iWAT) are moving into the limelight of fundamental research and therapeutic pursuits (**Sun et al., 2021**). This is spurred by the appreciation that human, just like rodent, white fat exerts important catabolic and endocrine functions (**Blondin, 2023**), begetting the seminal question about the nature of physiological, pharmacological, dietary, and lifestyle-associated interventions that will help achieve an increase in energy dissipation via scWAT. Here, the unsolved task is to prompt uncoupling protein 1 (UCP1)-dependent and UCP1-independent biochemical processes such as $Ca^{2+}$/creatine cycling (**Rahbani et al., 2021**; **Ikeda et al., 2017**), or by inducing an energy-consuming combination of triacylglycerol breakdown and fatty acid re-esterification that occurs after sympathomimetic administration (**Blondin et al., 2020**) selectively in beige fat. Intriguingly, a recent report demonstrated that oral supplementation of high-fat diet-fed mice with a nitroalkene derivative of salicylate (SANA) can induce creatine cycling in rodent iWAT independent of UCP1, demonstrating that the activation of iWAT thermogenesis and mitochondrial respiration is exploitable using pharmacology. Noteworthy, a Phase 1 study demonstrated that human volunteers receiving SANA exhibited a modest degree of weight loss, suggesting that strategies for iWAT mobilization in rodents might translate to (obese) patients (**Cal et al., 2025**). To date, orthogonal nutritional approaches for iWAT activation in mice rely on negative energy balances and include calorie restriction paradigms such as intermittent fasting (**Li et al., 2017**), yet conclusive data on their effect in human scWAT remains limited (**Roth et al., 2021**; **Plummer and Johnson, 2022**) and might incur detrimental processes such as the unwanted mobilization of lean (muscle and bone) mass. The selective omission or removal of methionine and cysteine has proven to be efficient in activating EE via iWAT thermogenesis and involves both humoral (FGF21 secretion) and sympathetic nervous system-dependent, UCP1-independent responses (**Lee et al., 2025**; **Varghese et al., 2025**; **Ghosh et al., 2017**). Future studies with extended durations should clarify whether run-in dietary priming, for example via dietary MetR, can enhance adaptive thermogenic responses during prolonged cold exposure, ideally monitored by continuous measurement of core body temperature and more in-depth molecular analysis of the browning phenotype in iWAT.

An intriguing opportunity for dietary MetR might lie in maximizing treatment effects against more clinical conditions ranging from cancer (**Gao et al., 2019**; **Sanderson et al., 2019**) to obesity. Here, drugs that mimic the action of the gut hormone glucagon-like peptide-1 (GLP1), that is so-called GLP1 receptor agonists (GLP1Ra) such as semaglutide, have radically transformed obesity treatment and reduce body weight via reducing appetite (**Müller et al., 2022**). However, to date, these drugs have the unfortunate downside of unfavorably reducing EE (**Löffler et al., 2021**). As lead-in calorie restriction allows mice to maintain EE and thus enhance the degree of achievable weight loss (**Petersen et al., 2024**), nutritional approaches might help to break the efficacy plateau that continues to plague GLP1Ras, collectively highlighting the notion that dietary interventions such as selective amino acid restriction prior or during pharmacological treatment deserve further experimental exploration and clinical testing.

Classically, DIT is synonymous with the thermic effect of food (TEF) and describes the postprandial rise in energy expenditure attributable to digestion, absorption, and metabolic processing

of nutrients (*Calcagno et al., 2019*). This canonical, meal-linked DIT is transient and scales with meal energy composition (*Swaminathan et al., 1985*). In contrast, dietary MetR elevates EE independent of acute feeding by engaging an endocrine-neuronal axis, where hepatic amino acid sensing activates the integrated stress response, increases FGF21 expression and circulating levels, and drives thermogenesis in WAT/BAT and a state of metabolic inefficiency, thereby increasing EE (*Fang et al., 2022*; *Forney et al., 2020*; *Varghese et al., 2025*). Related protein-restriction paradigms similarly raise EE through FGF21-mediated thermogenic programs (*Laeger et al., 2014*). Our and previous data demonstrate that MetR increases EE at thermoneutral-adjacent conditions and elicits depot-specific thermogenic transcriptional programs, even when benchmarked against cold exposure. We propose that MetR represents a true form of 'diet-induced thermogenesis' in a mechanistic sense that is a diet composition-initiated, hormone-driven thermogenic state that is distinct from the classic meal-processing TEF.

Altogether, our study provides important initial insights into depot-specific transcriptional responses to MetR, CE, and their combination, yet several limitations remain. The relatively short duration (7 days MetR, 24 hr CE) was sufficient for assessing acute transcriptional interactions but limited our ability to investigate long-term physiological adaptations or sustained metabolic remodeling. Furthermore, applying complementary omics analyses, including proteomics and metabolomics or flux analysis, would provide deeper mechanistic insights into metabolic interactions at the protein and metabolite levels.

## Materials and methods

All animal experiments were performed in accordance with the Directive 2010/63/EU from the European Union and approved by the Ministry of Environment and Agriculture Denmark (Miljø- og Fødevarestyrelsen) under license no. 2018-15-0201-01544 and reported in line with ARRIVE 2.0 guidelines.

### Animal husbandry and experiments

All experiments were performed in male mice on a C57BL/6 N (RRID:MGI:2159965) background (Taconic, Denmark). All mice were housed under a 12 hr light/dark cycle (lights on at 06:00, lights off at 18:00) in a temperature and humidity-controlled facility (22 ± 1°C) and had ad libitum access to diets and drinking water. 8/9-week-old mice were acclimatized to our animal facility on a chow diet (NIH-31, Zeigler Brothers Inc, 8% calories from fat) and housed in groups of three to four animals per cage for the habituation period. For reproducibility reasons and to avoid biases by 'social thermogenesis', we single-housed all mice for the duration of the experiments and concluded the experiments with mice being 16 weeks of age.

In study 1, we performed indirect calorimetry to assess physiological effects and interactions between diets and ambient temperature. For this, 15 mice were split randomly into three groups and fed cysteine-depleted diets containing either 0.8% Methionine (Control; Ctrl), 0.12% Methionine (Methionine-restricted; MetR) or 2% Methionine (Methionine-supplemented; MetS) for 6 days at 22°C and 5 days at 4°C. Diets were from Research Diets (New Brunswick, NJ, USA). Exact compositions can be retrieved under cat. no. A11051301B (MetR), A11051302B (Ctrl), and A21060801 (MetS).

In study 2, we performed detailed analyses of physiological (body and organ weights) and molecular parameters (RNA-seq, colorimetric/enzymatic assays) to investigate the molecular adaptations and interactions between diet and temperature. For this, 28 mice were split randomly into four groups and were either fed previously mentioned Ctrl or MetR diets at 22°C for 7 days or housed an additional 8th day at 4°C. Blood glucose levels were determined just prior to sacrifice in the cage using a handheld glucometer. The mice were then euthanized by carbon dioxide asphyxiation followed by cervical dislocation. Blood was collected by cardiac puncture and stored on ice for the duration of the sacrifice. Serum was collected after centrifugation by 2000 ×g for 15 min and stored at –80°C. Liver, eWAT, iWAT, and BAT were weighed and snap-frozen in liquid nitrogen and then stored at –80°C. All sacrifices were performed during the light phase between 09:00 and 11:00. Investigators were not blinded during animal handling and tissue collection. Downstream molecular and computational analyses were performed using coded sample identifiers to minimize bias. Sample sizes were selected based on prior experience with comparable metabolic and transcriptomic studies and were sufficient

to detect biologically meaningful differences. No formal a priori power calculation was performed. No animals were excluded from physiological analyses.

## Indirect calorimetry

Indirect calorimetry was conducted using the PhenoMaster NG 2.0 Home Cage System (TSE systems, Bad Homburg, Germany). Prior to the experiment, a complete calibration protocol for the gas analyzers was run according to the manufacturer's recommendations, and the mice were weighed. Mice were housed individually and acclimated to the new environment for 3 days prior to the experiment. The machine was set to maintain 50% humidity throughout the experiment and a 12 hr light/dark cycle, with ad libitum access to food and water. During the experiment, energy expenditure, respiratory exchange ratio, food and water intake were recorded every 60 s and datapoints were filtered for outliers and then averaged per hour for the analysis. Indirect calorimetry data are presented as mean ± SEM.

## RNA sequencing and analysis

Total RNAs from tissues were isolated using the TRI reagent (Sigma) followed by clean-up with RPE buffer (Qiagen, Germany). The quality of RNA was validated by the Agilent RNA 6000 Nano-Kit in Agilent 2100 Bioanalyzer according to the manufacturer's protocol (Agilent Technologies, Waldbronn, Germany). Only samples with RNA integrity number (RIN)≥8.5 were used for library preparation. mRNA sequencing was performed in-house. After quality control, 500 ng RNA in a final volume of 25 µL DEPC-treated water was prepared and sample preparation was performed as described in the NEBNext Poly(A) mRNA Magnetic Isolation Module kit and NEBNext Ultra II RNA Library Prep Kit (cat no: #E7770, New England Biolabs, Ipswich, MA, USA). The amplified libraries were validated by Agilent 2100 Bioanalyzer using a DNA 1000 kit (Agilent Technologies, Inc, Santa Clara, CA, USA) and quantified by qPCR using the KaPa Library Kits (KaPa Biosystems, Wilmington, MA, USA). Hereafter, 2×50 bp paired-end sequencing was performed on Illumina Novaseq 6000. Sequencing targeted approximately 20 million paired-end reads per sample. Seven samples per group and tissue were submitted to sequencing. One sample in iWAT and iBAT was lost during the sample preparation.

## RNAseq data analysis

Sequencing reads were aligned to the mouse reference genome mm10/GRCm38 (GENCODE vM25; RRID:SCR_014966; *Mudge et al., 2025*) using STAR aligner (v2.7.9a; RRID:SCR_004463; *Dobin et al., 2013*) with the following parameters: `--outSAMunmapped Within`, `--outFilterType BySJout`, `--outSAMattributes NH HI AS NM MD`, `--outFilterMultimapNmax 10`, `--outFilter-MismatchNoverReadLmax 0.04`, `--alignIntronMin 20`, `--alignIntronMax 1000000`, `--alignMatesGapMax 1000000`, `--alignSJoverhangMin 8`, and `--alignSJDBoverhangMin 1`. Gene-level quantification was performed using featureCounts (v2.0.3; RRID:SCR_012919; *Liao et al., 2014*). In parallel, transcript-level quantification was performed using Salmon (v1.9.0; *Patro et al., 2017*) and Gencode vM25, with parameters `--seqBias`, `--useVBOpt`, and `--numBoot-straps 30`. Transcript abundance estimates were imported and summarized to the gene level using the tximport package (*Soneson et al., 2015*). Quality control was summarized with MultiQC v1.19 (RRID:SCR_014982). All RNA-seq samples passed the QC. Differential gene expression analysis was performed using DESeq2 (v1.46.0; RRID:SCR_000154; *Love et al., 2014*) with shrinkage of fold changes using apeglm (*Zhu et al., 2019*). Genes with fewer than 10 total counts across all samples were excluded prior to normalization. The following interaction model was applied: ~ diet + temperature + diet:temperature, allowing detection of diet effects, temperature effects, and their statistical interaction. Specifically, the model tested: (i) the main effect of diet (MetR vs. Ctrl), (ii) the main effect of temperature (CE vs. RT), and (iii) the diet × temperature interaction (whether the effect of MetR differs between RT and CE). Genes with an adjusted p-value <0.05 and an absolute $\log_2$FC ≥0.585 were considered significantly differentially expressed. Gene set enrichment analysis (GSEA; RRID:SCR_003199) was performed using the gseKEGG() function from the clusterProfiler package (v4.14.6; RRID:SCR_016884; *Yu et al., 2012*) with parameters: organism = "mmu", pvalueCutoff =1, pAdjustMethod = "BH", minGSSize = 0, seed = TRUE, and eps = 0. Quality control metrics and preprocessing approaches were done as previously described (*Davidsen et al., 2024*).

## RNAseq outlier detection

Outlier detection was performed upon visual inspection of PCA plots within tissue analyses and was supported by robustPCA from the rrcov package (v1.7.7) (*Todorov, 2004*). In the order of Ctrl_RT, Ctrl_CE, MetR_RT, and MetR_CE, the following number of samples were excluded in the transcriptomic analysis: 0, 0, 0, and 0 (Liver); 0, 1, 0, and 0 (iBAT); 0, 0, 0, and 0 (iWAT); and 3, 0, 1, and 0 (eWAT).

## Classification of transcriptional interaction

Differentially expressed genes were classified according to their regulation across single and combined stimuli using a custom R script adapted from *Pantaleón García et al., 2022*. The four relevant contrasts were defined as Ctrl_CE vs Ctrl_RT (A), MetR_RT vs Ctrl_RT (B), MetR_CE vs Ctrl_RT (AB), and the interaction term (A×B). Only genes significant (padj <0.05 and |FC| > 1.5) in at least one comparison were included. For each gene, the direction of regulation (Up, Down, Unchanged) was assigned for every contrast. Genes were then first classified as cooperative when both single exposures (A and B) changed in the same direction or remained unchanged, or as competitive when they differed (e.g. one Up, one Down, or only one significant). Based on these combinations, genes were grouped into distinct regulation categories: co-dependent (significant only in AB but not in A or B), concordant (both A and B regulated in the same direction and AB responded similarly), discordant (A and B regulated in the same direction but AB changed oppositely), Ctrl_CE- or MetR_RT-dominant (competitive single exposures where AB followed A or B), and masked (*Figure 3D*). Masked genes were defined as significant in one or both single exposures but not in AB: Ctrl_CE-masked (significant only in A), MetR_RT-masked (significant only in B), or mutually masked (significant in both A and B but lost in AB, indicating opposing regulation under the combined stimulus; *Figure 3—figure supplement 1C*). To evaluate synergy, an expected additive effect was computed as the sum of $log_2$ fold changes from the single exposures (expected{A+B} = $log_2FC${A} + $log_2FC${B}). The synergy score was calculated as $log_2$ of the absolute ratio between the observed combined effect ($log_2FC${AB}) and the expected additive effect, that is $log_2|log_2FC${AB}/expected{A+B}|. When this expected additive effect was approximately zero, synergy was considered undefined for that gene. Genes without significant interaction effects were classified as 'additive'. Genes with significant interaction effects were further assigned as 'synergistic' (synergy score >0) or 'antagonistic' (synergy score <0), reflecting whether the observed combined effect surpassed or fell short of additivity. The regulation patterns are exemplified in *Figure 3D*, *Figure 3—figure supplement 1C*.

## Serum measurements

Serum NEFAs were determined using the NEFA-HR kit (cat. no. 434–91795 and 436–91995; FUJIFILM Wako Chemicals Europe GmbH). Serum triglycerides were determined using the LabAssay Triglyceride kit (cat no. 291–94501; FUJIFILM Wako Chemicals Europe GmbH). Serum β-hydroxybutyrate levels were determined using the β-HBA kit (cat. no. 2940; Instruchemie, Delfzijl, the Netherlands). Serum FGF21 and IL-6 levels were determined using the Luminex Multiplex platform (R&D Systems).

## Statistics

Exact n values and statistical tests are reported in the figure legends. Data are presented as mean ± SEM unless stated otherwise. Statistical analysis of the transcriptomics data was performed as described in the previous paragraph. Serum, body, and organ weight data were analyzed using GraphPad Prism (v. 10.4.1; RRID:SCR_002798). Prior to the analysis, the assumptions of normality and homogeneity of variance were assessed. These assumptions were met in all presented parameters. Plasma parameters, absolute and relative organ weights, and bodyweight change (%) were analyzed using an unpaired one-way ANOVA with Tukey's HSD test to correct for multiple comparisons. Statistical differences between the groups were depicted in a compact letter display, where groups sharing the same letter are not significantly different. Bodyweight data (start/finished) and average daily locomotor activity and food and water intake (RT/CE) were analyzed using a paired two-way ANOVA with Tukey's HSD test to correct for multiple comparisons. Statistical differences between the groups were depicted as * $p<0.05$, ** $p<0.01$, *** $p<0.001$, **** $p<0.0001$. Energy expenditure, food and water intake data from the indirect calorimetry experiment was analyzed by ANCOVA with body weight or locomotor activity as covariate. The ANCOVA analysis was done pairwise using the regression tool by NIDDK Mouse Metabolic Phenotyping Centers (MMPC, https://www.mmpc.org).

## Acknowledgements

We thank all the members of the Kornfeld lab for fruitful discussions. We gratefully acknowledge the assistance of the FGM-seq team for their help with the sequencing measurements and Victor Goitea for helpful discussions around outlier detection methods of RNA-seq samples. All of the computation done for this project was performed on the UCloud interactive HPC system, which is managed by the eScience Center at the University of Southern Denmark. This work was supported by an EMBO Long-Term Fellowship (ALTF 676-2021) to PMMR and a Novo Nordisk Foundation call NNF21OC0070263 to PMMR and JWK.

## Additional information

### Funding

| Funder | Grant reference number | Author |
| --- | --- | --- |
| European Molecular Biology Organization | ALTF 676-2021 | Philip MM Ruppert |
| Novo Nordisk Fonden | NNF21OC0070263 | Philip MM Ruppert Jan-Wilhelm Kornfeld |

The funders had no role in study design, data collection and interpretation, or the decision to submit the work for publication.

### Author contributions

Philip MM Ruppert, Conceptualization, Data curation, Software, Formal analysis, Supervision, Funding acquisition, Validation, Investigation, Visualization, Methodology, Writing – original draft, Project administration, Writing – review and editing; Aylin S Gueller, Resources, Formal analysis, Investigation; Marcus Skjæveland, Formal analysis, Investigation; Natasa Stanic, Resources, Investigation; Jan-Wilhelm Kornfeld, Conceptualization, Resources, Supervision, Funding acquisition, Validation, Methodology, Project administration, Writing – review and editing

### Author ORCIDs

Philip MM Ruppert ⓘ https://orcid.org/0000-0002-4028-8200
Aylin S Gueller ⓘ https://orcid.org/0009-0000-3636-2773
Marcus Skjæveland ⓘ https://orcid.org/0009-0009-7393-4721
Jan-Wilhelm Kornfeld ⓘ https://orcid.org/0000-0002-6802-4442

### Ethics

All animal experiments were performed in accordance with the Directive 2010/63/EU from the European Union and approved by the Ministry of Environment and Agriculture Denmark (Miljø- og Fødevarestyrelsen) under license no. 2018-15-0201-01544.

Reviewer #1 (Public review): https://doi.org/10.7554/eLife.108825.3.sa1
Reviewer #2 (Public review): https://doi.org/10.7554/eLife.108825.3.sa2
Reviewer #3 (Public review): https://doi.org/10.7554/eLife.108825.3.sa3
Author response https://doi.org/10.7554/eLife.108825.3.sa4

## Additional files

### Supplementary files

Source data 1. List of all differentially expressed genes.

Source data 2. List of all Gene set enrichment analyses.

MDAR checklist

## Data availability

The RNA-seq data generated in this study have been deposited in the NCBI Gene Expression Omnibus (GEO) under accession number GSE310044. The dataset includes raw FASTQ files, processed count matrices, and metadata for all experimental conditions. In addition lists of all DEGs and GSEA analysis from each tissue were provided in the supplementary files. Custom scripts and processed data required to reproduce the transcriptomic analyses and figures are publicly available via Zenodo https://doi.org/10.5281/zenodo.19024718.

The following datasets were generated:

| Author(s) | Year | Dataset title | Dataset URL | Database and Identifier |
|---|---|---|---|---|
| Ruppert PMM, Güller AS, Rosendal M, Stanic N, Kornfeld J | 2025 | Dietary sulfur amino acid restriction elicits a cold-like transcriptional response in inguinal but not epididymal white adipose tissue of male mice | https://www.ncbi.nlm.nih.gov/geo/query/acc.cgi?acc=GSE310044 | NCBI Gene Expression Omnibus, GSE310044 |
| Ruppert P | 2026 | MetRvsCE-transcriptomics | https://doi.org/10.5281/zenodo.19024718 | Zenodo, 10.5281/zenodo.19024718 |

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
