## [Editor Report · eLife Assessment]

The present study employed transcriptomics to investigate the impact of methionine restriction (MR) and cold exposure (CE) on liver and adipose tissues in mice. The authors demonstrate that responses to MR and CE are tissue-specific, while both MR and CE have a similar effect on beige adipose tissue. While these findings are somewhat descriptive, this work is considered **important**, as it provides a comprehensive resource for enhancing our understanding of these lifestyle interventions. The study is of high scientific quality, and the analyses are **convincing**.

---

## [Referee Report · Reviewer #1 (Public review)]

Summary:

Activation of thermogenesis by cold exposure and dietary protein restriction are two lifestyle changes that impact health in humans and lead to weight loss in model organisms, here the mouse. How these affect liver and adipose tissues has not been thoroughly investigated side by side. In mice, the authors show that the responses to methionine restriction and cold exposure are tissue-specific while the effects on beige adipose are somewhat similar.

Strengths:

The strength of the work is the comparative approach, using transcriptomics and bioinformatic analyses to investigate the tissue-specific impact. The work was performed in mouse models and is state-of-the-art. This represents an important resource for researchers in the field of protein restriction and thermogenesis.

Weaknesses:

The findings are descriptive and the conclusions remain associative. The work is limited to mouse physiology and the human implications have not been investigated yet.

---

## [Referee Report · Reviewer #2 (Public review)]

Summary:

This study provides a library of RNA sequencing analysis from brown fat, liver and white fat of mice treated with two stressors - cold challenge and methionine restriction - alone and in combination (interaction between diet and temperature). They characterize the physiologic response of the mice to the stressors, including effects on weight, food intake and metabolism. This paper provides evidence that while both stressors increase energy expenditure, there are complex tissue-specific responses in gene expression, with additive, synergistic and antagonistic responses seen in different tissues.

Strengths:

The study design and implementation is solid and well-controlled. Their writing is clear and concise. The authors do an admirable job of distilling the complex transcriptome data into digestible information for presentation in the paper. Most importantly, they do not over reach in their interpretation of their genomic data, keeping their conclusions appropriately tied to the data presented. The discussion is well thought out addresses some interesting points raised by their results.

Weaknesses:

The major weakness of the paper is the almost complete reliance on RNA sequencing data, but it is presented as a transcriptomic resource.

---

## [Referee Report · Reviewer #3 (Public review)]

Summary:

Ruppert et al. present a well-designed 2×2 factorial study directly comparing methionine restriction (MetR) and cold exposure (CE) across liver, iBAT, iWAT, and eWAT, integrating physiology with tissue-resolved RNA-seq. This approach allows a rigorous assessment of where dietary and environmental stimuli act additively, synergistically, or antagonistically. Physiologically, MetR progressively increases energy expenditure (EE) at 22{degree sign}C and lowers RER, indicating a lipid utilization bias. By contrast, a 24-hour 4 {degree sign}C challenge elevates EE across all groups and eliminates MetR-Ctrl differences. Notably, changes in food intake and activity do not explain the MetR effect at room temperature.

Strengths:

The data convincingly support the central claim: MetR enhances EE and shifts fuel preference to lipids at thermoneutrality, while CE drives robust EE increases regardless of diet and attenuates MetR-driven differences. Transcriptomic analysis reveals tissue-specific responses, with additive signatures in iWAT and CE-dominant effects in iBAT. The inclusion of explicit diet×temperature interaction modeling and GSEA provides a valuable transcriptomic resource for the field.

Comments on revisions:

The authors have addressed any concerns I had.

---

## [Author Response]

The following is the authors’ response to the original reviews.

**Reviewer #1 (Public review):**
Summary:Activation of thermogenesis by cold exposure and dietary protein restriction are two lifestyle changes that impact health in humans and lead to weight loss in model organisms - here, in mice. How these affect liver and adipose tissues has not been thoroughly investigated side by side. In mice, the authors show that the responses to methionine restriction and cold exposure are tissue-specific, while the effects on beige adipose are somewhat similar.Strengths:The strength of the work is the comparative approach, using transcriptomics and bioinformatic analyses to investigate the tissue-specific impact. The work was performed in mouse models and is state-of-the-art. This represents an important resource for researchers in the field of protein restriction and thermogenesis.Weaknesses:The findings are descriptive, and the conclusions remain associative. The work is limited to mouse physiology, and the human implications have not been investigated yet.

We thank Reviewer 1 for their thoughtful review and for highlighting the strength of our comparative, tissue-specific analyses. We acknowledge that our study is descriptive and limited to mouse physiology, and agree that translation to humans will be an important next step. By making these data broadly accessible, we aim to provide a useful resource for future mechanistic and translational studies on dietary amino acid restriction and thermogenesis.

**Reviewer #2 (Public review):**
Summary:This study provides a library of RNA sequencing analysis from brown fat, liver, and white fat of mice treated with two stressors - cold challenge and methionine restriction - alone and in combination (interaction between diet and temperature). They characterize the physiologic response of the mice to the stressors, including effects on weight, food intake, and metabolism. This paper provides evidence that while both stressors increase energy expenditure, there are complex tissue-specific responses in gene expression, with additive, synergistic, and antagonistic responses seen in different tissues.Strengths:The study design and implementation are solid and well-controlled. Their writing is clear and concise. The authors do an admirable job of distilling the complex transcriptome data into digestible information for presentation in the paper. Most importantly, they do not overreach in their interpretation of their genomic data, keeping their conclusions appropriately tied to the data presented. The discussion is well thought out and addresses some interesting points raised by their results.Weaknesses:The major weakness of the paper is the almost complete reliance on RNA sequencing data, but it is presented as a transcriptomic resource.

We thank Reviewer 2 for their positive evaluation of our study and for highlighting the strengths of our design, analyses, and interpretation. We acknowledge the limitation of relying primarily on RNA-seq, and emphasize that our intent was to provide a comprehensive transcriptomic resource to guide future mechanistic work by the community.

**Reviewer #3 (Public review):**
Summary:Ruppert et al. present a well-designed 2×2 factorial study directly comparing methionine restriction (MetR) and cold exposure (CE) across liver, iBAT, iWAT, and eWAT, integrating physiology with tissue-resolved RNA-seq. This approach allows a rigorous assessment of where dietary and environmental stimuli act additively, synergistically, or antagonistically. Physiologically, MetR progressively increases energy expenditure (EE) at 22{degree sign}C and lowers RER, indicating a lipid utilization bias. By contrast, a 24-hour 4 {degree sign}C challenge elevates EE across all groups and eliminates MetR-Ctrl differences. Notably, changes in food intake and activity do not explain the MetR effect at room temperature.Strengths:The data convincingly support the central claim: MetR enhances EE and shifts fuel preference to lipids at thermoneutrality, while CE drives robust EE increases regardless of diet and attenuates MetR-driven differences. Transcriptomic analysis reveals tissue-specific responses, with additive signatures in iWAT and CE-dominant effects in iBAT. The inclusion of explicit diet×temperature interaction modeling and GSEA provides a valuable transcriptomic resource for the field.Weaknesses:Limitations include the short intervention windows (7 d MetR, 24 h CE), use of male-only cohorts, and reliance on transcriptomics without complementary proteomic, metabolomic, or functional validation. Greater mechanistic depth, especially at the level of WAT thermogenic function, would strengthen the conclusions.

We thank Reviewer 3 for their thorough review and for recognizing the strengths of our factorial design, physiological assessments, and transcriptomic analyses. We acknowledge the limitations of short intervention windows, male-only cohorts, and the reliance on transcriptomics. Our aim was to generate a well-controlled comparative dataset as a resource, and we agree that future work incorporating longer interventions, both sexes, and additional mechanistic layers will be important to build on these findings.

**Reviewer #1 (Recommendations for the authors):**
In my opinion, the comparative analysis between tissues and treatments could be expanded.

We thank the reviewer for this suggestion. We included top30 DEG heatmaps for the comparison MetR_CEvsCtrl_RT for up and downregulated genes in the figures for each tissue. We also provide additional data in the supplementary, including top30 heatmaps for Ctrl_CEvsCtrl_RT, MetR_RTvsCtrl_RT, the interaction term, as well as one excel sheet per tissue for all DEGs (p<0.05 and FC +/- 1.5 and for all gene sets, GSEA).

**Reviewer #3 (Recommendations for the authors):**
(1) CE robustly increases food intake, yet MetR mice at room temperature, despite elevated EE, do not appear to increase feeding to maintain energy balance. The authors should discuss this discrepancy, as it represents an intriguing avenue for follow-up.

See answer below.

(2) CE raises EE to ~0.9 kcal/h irrespective of diet, suggesting that the additive weight loss seen with MetR+CE (Fig. 1H) must be due to reduced intake. This raises the possibility that MetR mice fail to appropriately sense negative energy balance, even under CE, and do not compensate with higher feeding.

We thank the reviewer for comments 1 and 2. We did not put an emphasis on this finding, as the literature on the effects on food intake under sulfur amino acid restriction are very inconsistent. Intial studies (e.g. by Gettys group) most often report on food intake per gram bodyweight and report an increase in caloric intake. We think that this reporting is flawed and should rather be reported as cumulative food intake. The recent paper by the Dixit group also reports that there is no effect on food intake, in line with our data. The recent paper by the Nudler group reports a decrease in food intake.

(3) Report effect sizes and sample sizes alongside p-values in all figure panels, and ensure the GEO accession (currently listed as "GSEXXXXXX") is provided.

We thank the reviewer for noticing this. So far we were unable to upload the datasets to GEO. We’re unable to connect to the NIH servers, presumably due to the US government shutdown. We are commited to sharing this dataset as soon as possible and will update the manuscript in the future accordingly. We included the sample size for experiment 1 and 2 in the figure legends and described our outlier detection method in the methods section. Significances are explained in the figure legends.

(4) Explicitly define the criteria for "additive," "synergistic," and "antagonistic" interactions (both at the gene and pathway levels) to help readers align the text with the figures.

We thank the reviewer for this helpful comment. We added an description of how we defined and computed the regulatory logic in the method section.

(5) Revise the introduction to address recent data from the Dixit group (ref. #38), which shows that EE induced by cysteine restriction and weight loss is independent of FGF21 and UCP1. As written, the introduction states: "Recent studies have shown that DIT via dietary MetR augments energy expenditure in a UCP1-dependent...fashion".

See answer below.

(6) "Mechanistically, MetR...results in secretion of FGF21. In turn, FGF21 augments EE by activating UCP1-driven thermogenesis in brown adipose tissue via β-adrenergic signaling (4,7)." This should be updated for accuracy and balance.

We thank the reviewers for both comments 5 and 6. Both recent publications by the Dixit and the Nudler groups (now ref 9 and 10) provide very interesting further mechanistic detail into the bodyweight loss in response to dietary sulfur amino acid restriction. However, there are also older papers by the Gettys group that in part contradict their findings, particularly, when it comes to the importance of UCP1 for the adaptation to sulfur amino acid restriction. Overall, we think that further work is required to determine the importance of UCP1-driven EE from alternative mechanisms that ultimately drive body and fat mass loss. We rewrote the referenced paragraph in the introduction to reflect this.